# Viral Vector-Based Melanoma Gene Therapy

**DOI:** 10.3390/biomedicines8030060

**Published:** 2020-03-16

**Authors:** Altijana Hromic-Jahjefendic, Kenneth Lundstrom

**Affiliations:** 1Department of Genetics and Bioengineering, Faculty of Engineering and Natural Sciences, International University of Sarajevo, 71000 Sarajevo, Bosnia and Herzegovina; ahromic@ius.edu.ba; 2PanTherapeutics, CH 1095 Lutry, Switzerland

**Keywords:** melanoma, cancer, vector delivery, gene therapy, immunotherapy, clinical trials

## Abstract

Gene therapy applications of oncolytic viruses represent an attractive alternative for cancer treatment. A broad range of oncolytic viruses, including adenoviruses, adeno-associated viruses, alphaviruses, herpes simplex viruses, retroviruses, lentiviruses, rhabdoviruses, reoviruses, measles virus, Newcastle disease virus, picornaviruses and poxviruses, have been used in diverse preclinical and clinical studies for the treatment of various diseases, including colon, head-and-neck, prostate and breast cancer as well as squamous cell carcinoma and glioma. The majority of studies have focused on immunotherapy and several drugs based on viral vectors have been approved. However, gene therapy for malignant melanoma based on viral vectors has not been utilized to its full potential yet. This review represents a summary of the achievements of preclinical and clinical studies using viral vectors, with the focus on malignant melanoma.

## 1. Introduction

Melanoma, or malignant melanoma, represents a cancer type that develops in melanocytes known as pigment-containing cells [1]. Since the beginning of the 21st century, melanoma has remained one of the most fatal malignancies. Most patients, when diagnosed early, are treated by local surgical excision following sentinel lymph node biopsy [1]. The incidence varies by country, skin phenotype and sun exposure. It mostly affects young and middle-aged female populations (below the age of 50 years), but more males are affected from the age of 55 onwards [1]. In men the incidence of melanoma is three times higher than in women by the age of 75. Ultraviolet (UV) light is known to be the main cause of malignant melanoma. A history of sunburn in childhood or adolescence has been suggested to be directly associated with the development of melanoma. Other risk factors include the number of melanocytic nevi, family history and genetic background. It has also been confirmed that patients with a previous history of melanoma are more prone to develop multiple primary melanomas [1]. In contrast, other environmental factors like alcohol or tobacco consumption have not been associated with melanoma development [2]. 

Skin melanoma has generally been classified according to the origin of the sun exposure, the degree of cumulative UV exposure, age at the time of diagnosis, types of oncogenic drivers and the mutational load [2]. It is known that B-Raf proto-oncogene (BRAF), neurofibromin 1 (NF1) and NRAS mutations, together with a high mutational load related to UV exposure, are the main genetic drivers [3]. Cases of periodic sun exposure are usually associated with BRAF^V600E^ and a lower mutational load [2]. It is important to mention that each melanoma subtype may evolve from different precursor lesions, which can involve different gene mutations as well as different transformational stages [3].

Current medical treatments include various methods. Most patients with recently diagnosed melanoma have early-stage disease and can be treated by surgical excision, which is curative in the majority of cases. Some treatment methods involve lymph node biopsies in addition to standard surgical excision. Unfortunately, 10% of all melanoma cases are diagnosed at an advanced/late stage and are already metastatic, including visceral and brain metastases [2,3,4,5]. These patients have a poor prognosis and the probability of treatment success is lower. For patients with advanced stage disease, revolutionary therapy agents including RAF (Rapidly Accelerated Fibrosarcoma) and MEK (Mitogen-activated Protein Kinase) kinase inhibitors as well as immune checkpoint inhibitors like anti-CTLA4 and anti-PD1 have been approved, in 2011 and 2016, respectively [5,6,7,8]. Anti-PD1 and anti-CTLA4 antibodies (nivolumab, pembrolizumab and ipilimumab) as well as BRAF and MEK inhibitors (vemurafenib and trametinib) have shown promising results in clinical trials [9,10,11,12,13,14,15,16]. Today, the presence of the BRAF^V600E^ mutation is verified in clinical settings, since it determines the correct treatment strategy. Mutations like NRAS, NF1, CKIT, CDKN2A and PTEN have not been included in clinical practice yet.

Immunotherapy and kinase inhibitors are known as backbones for second-line systemic chemotherapy [17]. In the past, chemotherapy represented the treatment option for advanced melanoma. Although attempts to improve patient responses by combination therapy failed, it is still used for palliative treatment of progressed melanomas [18]. Dacarbazine, an alkylating agent, was approved by the FDA in 1974 for standard chemotherapy treatment of metastatic melanoma [19]. Despite moderate results, dacarbazine has been used as the sole standard of care, recently (in clinical trials) in combination with other chemotherapies and immunotherapies (ClinicalTrials.gov) [20]. Temozolomide (TMZ), an active metabolite of dacarbazine, has been applied in advanced melanoma [20].

Electrochemotherapy (ECT) combines two cytotoxic drugs (cisplatin and bleomycin) with high-intensity electric pulses, which enhances the delivery of the drug into cells [21,22]. This approach has been used for the treatment of cutaneous and subcutaneous melanoma nodules [23]. The overall response was 85%, and no major negative adverse events were reported [21]. Photodynamic therapy (PDT) is a light-based therapy. It represents a promising adjuvant treatment and can be used as a palliative method of choice for patients with stage III/IV cutaneous metastatic melanomas [24]. PDT is considered a minimally invasive procedure that requires a photosensitizer. Absorption is superior in metabolically active tissues [24]. The method applies non-toxic compounds, which create reactive oxygen species (ROS) when combined with oxygen [25]. ROS does irreversible damage to tumor cells and tumor-associated blood vessels, and contributes to the activation of various antitumor, immune and inflammatory responses [25,26,27,28]. Although PDT can be applied to both nonmalignant and malignant diseases, several reports show that PDT alone has only limited efficacy in melanoma [22]. To improve PDT results, certain protective mechanisms, like pigmentation and oxidative stress resistance, need to be overcome [29,30]. The combination of PDT and dacarbazine chemotherapy displayed resistance reduction in pigmented and unpigmented metastatic melanomas [31]. On the other hand, combination of PDT and immunotherapy may increase the effect of eradication of the initial tumor and decrease in melanoma recurrence [29].

In the context of oncolytic viruses, their selective replication triggers tumor cell death and vector spread into neighboring cells, providing an interesting approach for cancer therapy [32]. Immunization studies in experimental models have employed a wide range of viral vectors based on adenoviruses, alphaviruses, herpes simplex viruses, coxsackie viruses and vaccinia viruses, targeting cancer types like glioblastoma, colon, cervix, and lung cancer as well as melanoma [33,34]. The first virus-based melanoma drug was approved by the FDA in 2015 [35]. 

## 2. Viral Vector Systems

Today, both viral and non-viral vectors have been subjected to preclinical and clinical studies in the field of gene therapy. A brief description of different viral vector systems for gene therapy applications is presented below, with a special emphasis on melanoma treatment. The range of viral vectors is very broad. It includes delivery vehicles developed for both short-term and long-term expression, including single-stranded (ss) and double-stranded (ds) RNA and DNA genome viruses [36]. The major features of the viral vectors, and examples of their applications in gene therapy, are presented in Table 1. 

## 3. Viral Vectors for Melanoma Treatment

As previously described, a broad range of oncolytic viruses have been evaluated for cancer gene therapy [107]. The specific targeting and killing of tumor cells and the simultaneous stimulation of the immune system have made oncolytic viruses attractive delivery vehicles [108,109,110]. This dual action promotes tumor regression as well as the induction of immune responses through innate and adaptive components. On the other hand, naturally occurring, ubiquitous, non-enveloped dsRNA viruses have shown generally mild infection in humans, and specific replication and cytopathogenicity in transformed cells, which possess active Ras signaling pathways [111,112]. Their specificity for Ras transformed cells and their relatively non-pathogenic nature in humans make them attractive anticancer therapy candidates [111,112]. This approach may lead to the recognition and removal of systemic disease and the prevention of tumor return [113]. A summary of completed and ongoing clinical trials is presented in Table 2. 

### 3.1. Melanoma Treatment Using Herpes Simplex Virus Type 1

The prototype drug for virotherapy is an attenuated herpes simplex virus type 1 (HSV-1), which is engineered to express the human granulocyte–macrophage colony-stimulating factor (GM–CSF) [35]. The approved drug known as talimogene laherparepvec (TVEC) has the trade name Imlygic^®^. TVEC showed two mechanisms of action, one being the oncolytic effect of infecting and killing tumor cells at the local injection site, and the other being the immunotherapeutic effect through induction of local and systemic immune responses [114]. 

TVEC replicates in tumor cells, which results in lysis and release of soluble tumor-associated antigens and viral pathogens. Migration and maturation of dendritic cells is induced by local GM-CSF expression, leading to ingestion of dissolvable tumor antigens and apoptotic tumor cells. The dendritic cells are transported to the nearest lymph nodes, where antigens initiate a systemic immune response, specifically in CD4+ and CD8+ helper and cytotoxic T-cells. However, the response rate in metastases is lower than in injected tumors, which reflects insufficient effector T-cell expansion. The other reason could be the lack of efficacy at distant sites. To overcome this limitation, combination therapy with TVEC and immune checkpoint blockers might provide better results [35]. 

Generally, local lytic TVEC infection in tumor cells leads to the release of various proteins, such as interferons, chemokines, danger-associated molecular pattern (DAMP), and pathogen-associated molecular pattern (PAMP). These can provide more favorable surroundings for stimulation of anti-tumor immune responses [115]. Cancer cell lysis discharges tumor-associated neoantigens for processing by dendritic cells, which are activated by TVEC-encoded GM–CSF. This could lead to stimulation of anti-tumor CD8+ T-cell responses against unrecognized antigens and this effect has been clinically demonstrated [115].

#### 3.1.1. Preclinical Studies with HSV

TVEC showed strong lytic activity against various human tumors, including malignant melanoma. Melanoma patients treated with TVEC exhibited accumulation of CD8+ T-cells in injected lesions and decrease of CD4+ FoxP3+ regulatory cells as well as CD14+ myeloid-derived suppressor cells [115]. Together, these findings demonstrate the effect of TVEC in the infection process and the driving of transgene expression in different cell types.

#### 3.1.2. Clinical Trials of TVEC in Melanoma Treatment

The ICP34.5-deleted HSV-1 vector was engineered to express beta-galactosidase [122]. Strong expression was observed in the brain and dorsal root ganglia, with only minor toxicity, which indicates that deletion of the HSV-1 neurovirulence gene eliminates neurotoxicity. However, it preserves tolerable infectivity, in order to allow dominant trans-gene expression in neural tissue [122]. The ICP34.5-deleted HSV-1 vector was more effective with respect to lysis in human breast cancer cells than in a mixed culture of autologous hematopoietic cells [123]. 

The first human study of TVEC was a phase I trial, involving 30 patients [123]. Patients were chosen according to cutaneous and subcutaneous metastases from melanoma, breast cancer or squamous cell carcinoma of the head and neck [123]. Local intratumoral administration of virus was applied in order to reduce toxicity compared to systemic delivery, and this was well tolerated with only few adverse effects, like local inflammation and flu-like symptoms. Local reactions were dose-limited (10^7^ pfu/mL) and baseline HSV serologic status did not influence the effect of TVEC. However, a priming dose of 10^6^ pfu/mL was established, followed 3 weeks later by increasing the dose to 10^8^ pfu/mL every 2 weeks until disease progression was confirmed or unacceptable toxicity occurred [123]. Although objective responses were not seen in the trial, viral replication as well as GM–CSF expression and HSV-antigen-associated necrosis were observed in melanoma, breast, head and neck cancer. Six patients manifested flattening of injected/un-injected tumors, and in four patients, systemic immune responses were obtained.

In a phase II trial, 50 patients with metastatic melanoma at stage IIIC or IV were enrolled [124]. Systemic treatment was carried out for 74% of the patients with advanced melanoma. Patients were injected with 10^6^ pfu/mL, and the dose was increased 3 weeks later to 10^8^ pfu/mL. Treatment continued every 2 weeks for a maximum of 24 treatments [124]. TVEC was again well tolerated and the toxicity was primarily narrowed to flu-like symptoms and local inflammation. The overall survival rate was 58% at one year and 52% at two years [9].

After encouraging results were obtained from the phase II trials, a phase III trial (OPTiM) was conducted for patients with stage IIIB to IV melanoma [117]. Patients either received intralesional TVEC or GM–CSF subcutaneously, at 125 mg/m^2^ daily for 14 days in 28-day cycles. Since the frequency of pseudo-progression with TVEC was high, treatment discontinuation was not required before 24 weeks. After this period, treatment continued. The primary endpoint was a durable response rate (DRR), defined as the percentage of patients experiencing a response lasting more than 6 months and beginning within the first 12 months of treatment. The most common adverse effects upon application of TVEC were fatigue, chills, nausea and local injection symptoms. Statistical analysis revealed that overall survival in the TVEC arm was superior for patients with stage III and IV M1a melanoma (metastases only in skin or lymph nodes) and for treatment of naive patients with *p* < 0.001 in both comparisons. According to these remarkable results, TVEC became the first oncolytic virus-based therapy which showed significant clinical benefits in a phase III trial [114] leading to the FDA approval of TVEC as a monotherapy in October 2015 [35].

### 3.2. Melanoma Treatment Using Retroviruses/Lentiviruses

Retroviruses and lentiviruses are ssRNA, which can provide long-term transgene expression by integration into the host genome. They have frequently been used as gene therapy vectors for indications such as glioma [77,78], and breast [81], gastric [82], liver [83], pancreatic [84], and hematologic [85] cancers. One limitation of using retroviruses such as Moloney murine leukemia virus (MoMLV) for gene therapy is the requirement of cell division for transduction and integration [125]. In contrast, lentiviruses are capable of transduction of both dividing and non-dividing cells.

#### 3.2.1. Preclinical Studies with Retroviruses/Lentiviruses

Although retroviruses have demonstrated potential for treating chronic diseases such as severe combined immunodeficiency (SCID) in children [126], fewer studies have been conducted for cancer. For instance, recombinant retrovirus vectors expressing GM–CSF and IL-4 showed high-level expression in cultured primary glioma cells, which lasted for 14 days and could therefore present an attractive approach for immunotherapy [77]. However, in recent years, lentiviral vectors have replaced conventional retroviruses in gene therapy. For instance, a lentivirus carrying the EGFP reporter gene provided long-term expression in DU145 and PC3 human prostate cell lines and in vivo in pre-established and orthotopic tumors [127]. In the context of melanoma, a lentiviral vector expressing the VP22-CD/5-FC suicide gene system demonstrated superior antitumor activity in a murine uveal melanoma model [128]. In another study, a lentivirus vector expressing RNAi sequences targeting the MAT2B gene, the regulatory subunit of methionine adenosyltransferase resulted in suppressed growth, colony formation and induced apoptosis in A375 and Mel-RM malignant melanoma cell lines, and affected tumor growth in a xenograft model in vivo [129]. Moreover, antisense non-coding mitochondrial RNA (ASncmtRNAs) was downregulated by a lentivirus vector expressing short hairpin RNA (shRNA), which induced apoptosis in murine B16F10 and human A375 melanoma cell lines, significantly reduced B16F10 tumor growth in vivo, and reduced the number of lung metastases in a tail vein assay [130]. 

#### 3.2.2. Clinical Trials of Retroviruses/Lentiviruses for Melanoma Treatment

Related to lentivirus-based clinical trials, 30 children and adults with relapsed acute lymphoblastic leukemia (ALL) were treated with a lentiviral vector-based chimeric antigen receptor T (CAR-T), targeting CD19 (CTL019), which resulted in sustained remission with a 6-month event-free survival rate of 67% and an overall survival rate of 78% [131]. The treatment of relapsed and refractory ALL was efficient, with a high remission rate lasting for up to 24 months. In preparation for lentivirus-based clinical trials, monocyte-derived conventional dendritic cells (ConvDCs) were transduced using a tricistronic lentivirus vector, expressing GM–CSF, IL-4 and the melanoma antigen tyrosine-related protein 2 (TRP2), to overcome the difficulties in manufacturing and potency of ConvDCs [132]. The feasibility of this approach was demonstrated with monocytes from five advanced melanoma patients indicating that a simpler GMP-compliant method for manufacturing individualized DC vaccines with a higher specificity against melanoma is possible. In another approach, to improve ex vivo manufacturing of engineered T cells, isolated human CD8+ T cells from healthy donors were transduced with a lentivirus vector expressing the gp100-specific tumor antigen-specific T cell receptor (TCR) in the presence of a novel chemical lentiviral transduction enhancer (Lentiboost) [133]. It was demonstrated that antigen-specific secretion of tumor necrosis factor (TNF) and interferon-γ (IFN-γ) occurred in the transduced cells and significant cytotoxicity was detected in the antigen-positive tumor cells, showing the potential of lentivirus-based cancer immunotherapy.

The success of CAR-T based lentivirus therapy for hematological cancers such as ALL has also triggered treatment of solid tumors [134]. However, with tumors, the transition might be limited by therapeutic barriers such as CAR-T cell expansion, persistence, trafficking, and fate. In the context of melanoma, the first results from CAR-T cell therapy could not reproduce the findings from the treatment of hematological diseases [135]. Issues to be addressed include the lack of migration of CAR-T cells from blood vessels to the tumor site, as well as the immunosuppressive tumor microenvironment within solid tumors, before this technology can be successfully applied for melanoma treatment.

### 3.3. Melanoma Treatment Using Reoviruses

The nonenveloped dsRNA Reovirus Serotype 3-Dearing Strain known as Reolysin has been shown to replicate in specifically transformed cells possessing an activated RAS signaling pathway, which makes it a potential candidate for anticancer therapy [136,137,138]. The inhibition of dsRNA-activated protein kinase (PKR) in Ras-activated cells inhibits autophosphorylation of PKR, permitting viral translation and oncolysis in tumor cells [36,114]. As Ras pathway activation occurs in approximately 60% of metastatic melanoma patients, it provides a great opportunity for Reolysin testing in malignancy treatment [139,140].

#### 3.3.1. Preclinical Studies with Reoviruses

It has been observed that some tumor cells and spontaneously transformed cell lines show favorable sensitivity toward reoviruses [141]. Other studies have revealed that intratumoral injection of reoviruses leads to regression of v-erbB-transformed NIH 3T3 or human U87 glioblastoma tumors in 80% of SCID mice. Moreover, multiple reovirus injections resulted in total tumor regression in 65% of immune-competent C3H mice [142]. The oncolytic capacity of reoviruses allows their application as single drugs targeting various types of cancers. It was confirmed that reoviruses can kill six human breast cancer cell lines (SK-BR-3, KPL4, MDA-MB-453, CRL1500, MCFT and MDA-MB-231) expressing HER-2, but not the breast cancer cell line Hs578Bst, which did not show HER-2 expression [143]. Favorable anticancer effects have also been demonstrated in immunocompetent mouse models in the presence of cyclosporine A or anti-CD4/anti-SD8 antibodies as immunosuppressant agents [144]. This shows the potential for reoviruses to be implemented for the treatment of a wide range of cancer types.

#### 3.3.2. Clinical Trials of Reoviruses in Melanoma Treatment

Reovirus-based monotherapy has been conducted for solid tumors including soft-tissue sarcomas, melanoma, breast cancer and head and neck cancer [145]. One study included 18 patients and was mainly designed to verify the safety and tolerability of reovirus intralesional administration [146]. Monitoring was done over a period of six weeks. The toxic effects were measured according to the criteria of the National Cancer Institute Clinical Trials Group. The tumor responses were measured using the Response Evaluation Criteria in Solid Tumors. After the trial period of six weeks, one patient showed complete response (CR), two demonstrated partial responses (PRs), four patients had stable disease (SD) and ten showed progressive disease (PD) [146]. However, the results indicated that intralesional reovirus monotherapy was safe, well tolerated and did not reach dose-limiting toxicity. Local administration of reovirus in phase I/II malignant melanoma studies was as well tolerated as monotherapy [147,148]. Intravenous reovirus monotherapy was applied in a malignant melanoma phase II trial in 21 patients receiving a 3 × 10^10^ 50% tissue-culture infective dose (TCID50) once every 60 min on days 1–5 every four weeks [73]. Clinical benefits as CR or PR were monitored for eight weeks. One patient demonstrated extensive tumor necrosis (75%–90%) in two metastatic lesions after two treatment cycles, whereas no other patient met the criteria for CR or PR [119]. Moreover, reovirus was detected in two out of 13 biopsies containing melanoma metastases. The findings from the phase II trial further support the positive clinical outcome obtained from phase I studies [149,150]. Furthermore, 2 out of 13 patients showed productive reoviral replication in melanoma metastases. Unfortunately, the trial could not progress as initially planned, as the clinical objective of having two or more patients reaching CR or PR was not achieved. According to these results, the phase II trial did not support the application of reovirus as a monotherapy for metastatic melanoma, but rather as part of a combination therapy with other therapeutic or chemotherapeutic agents [146]. 

### 3.4. Current Stage of Melanoma Treatment Using Coxsackievirus CVA21

CVA21, a member of the Picornaviridae family, is a nonenveloped ssRNA enterovirus enclosed in an icosahedral capsid. Two major subgroups, A and B, have been characterized in murine models [151]. Subgroup A contains 23 serotypes, with their main impact on skeletal muscles, while subgroup B contains six serotypes affecting a broad range of tissue types [149]. Their clinical significance in humans reflects their responsibility for mild upper respiratory tract infections spread by aerosol transmission [152]. The oncolytic CVA21 is commercially available as CAVATAK™ based on the wild-type Kuykendall strain [153]. Modelling of the attachment mechanisms and cell internalization has indicated that other group A serotypes may have similar oncolytic potential [154]. The CVA21 infection is characterized by attachment to the intracellular adhesion molecule-1 (ICAM-1), the primary receptor for attachment, and to the decay-accelerating factor (DAF), the secondary receptor for attachment [155]. ICAM-1 is a viral receptor common for the Picornaviridae family. Although DAF is expressed on almost all cells, its primary role is regulation of complement responses [155]. Since the attachment of CVA21 to DAF is not sufficient for host cell infection, DAF is considered to act as a membrane receptor, which accumulates the virus at the cell surface and optimizes viral entry via ICAM-1 [156]. The discovery of CVA21 and its lysis of cancer cells was mainly achieved through research conducted on ICAM-1 and DAF receptors, and comparisons between various cancer cell lines and nonmalignant tissues. It has also been shown for melanoma as well as multiple myeloma, malignant glioma, breast, colon, endometrial and pancreatic cancer cell lines [157,158]. 

#### 3.4.1. Preclinical Studies with CVA21 

Overexpression of ICAM-1 and DAF in malignant melanoma compared to normal cells represents the potential for CVA21 applications [159]. Upregulation of ICAM-1 in melanoma is well studied, and it has been considered a relevant marker in disease prognosis [160]. It has been suggested that ICAM-1 is responsible for metastasis generation, as it facilitates cell-cell interactions between malignant melanocytes and circulating lymphocytes [95]. Upregulation of ICAM-1 in melanoma could be a potential target for CVA21 therapy as has been reported [158]. These results were obtained from the human melanoma chemotherapy-resistant Mel-CV and Mel-FH cell lines. The primary melanoma MM200 cell line showed in vitro oncolysis [161]. In vivo, it was shown that mouse tumor xenografts were reduced after treatment with CVA21 through intratumoral, intravenous and intraperitoneal administration [86]. Later, it was demonstrated that other group A coxsackie viruses had similar oncolytic properties and were not sensitive to neutralizing antibodies against CVA21. This opens the possibility of multivalent approaches through the application of different serotypes [156]. 

#### 3.4.2. Clinical Trials Using CVA21 

Investigations of CVA21 have progressed to clinical trials; phase I safety trials were completed in 2009 [162]. Fifty-seven patients with unresectable stage IIIc-IVM1c melanoma were subjected to an intratumoral dose of 3 × 10^8^ TCID_50_ of CAVATAK™ on days 1, 3, 5 and 8 in a phase II trial. A phase II trial using CVA21 enrolled 57 patients with unresectable stage IIIc-IVM1c melanoma. The therapy continued every three weeks, with a total of six injections followed by sequential tumor biopsies for injected and non-injected lesions, where monitoring of viral replication and virus-induced immune activation was observed. The obtained DRRs were 28.1 % (16 out of 57 patients) and 38.6% (21 out of 57 patients) and showed immune-related progression-free survival six months after the treatment [163]. Additionally, the first results from phase Ib reported the combination therapy of CAV21 with ipilimumab in patients with advanced melanoma [151]. A cohort of 26 patients was chosen, where 13 individuals had previously been treated with anti-PD-1 antibody. All 26 of them received 3 × 10^8^ TCID_50_ CVA21 on days 1, 3, 5, 8 and 22, followed by six injections of ipilimumab (3 mg/kg), beginning on day 22. The ORR was 38% and the disease control rate (DCR) was 88%, including CR, PR and stable disease SD [164]. According to a phase II study, intralesional admission of CVA21 can substantially influence the dynamics of the tumor micro-environment, which can be observed by an increase in immune cell infiltrates and immune-related response genes [116]. 

### 3.5. Current Stage of Melanoma Treatment Using Newcastle Disease Virus (NDV)

The LaSota strain of NDV has been modified by reverse genetics in order to be applied as a recombinant vaccine vector [165]. The distinct advantages of NDV relate to the host restriction and lack of pre-existing immunity in humans, where high-titer NDV replication can be obtained in certain cell lines suitable for human vaccine development [166]. Reverse genetics was employed in order to optimize foreign gene expression from NDV vectors. This was achieved through application of red fluorescence protein (RFP) at multiple sites downstream of the nucleocapsid (NP), P, M, fusion (F), hemagglutinin-neuramidase (HN) and large polymerase (L) genes [167]. The sequential transcription mechanism was confirmed according to the RFP expression levels. In a similar way, the avirulent NDV strain TS09-C was modified using reverse genetics for GFP expression after gene insertion upstream of the NP, M and L genes, resulting in remarkable expression [168]. The specific replication in tumor cells makes NDV attractive for cancer therapy [89].

#### 3.5.1. Preclinical Studies with NDV

Regarding preclinical studies, the NDV strain 73-T caused complex tumor regression in athymic mice containing fibrosarcoma xenografts [169]. Even a single intratumoral administration caused complete regression and multiple administrations provided superior efficacy [169]. Intratumoral administration of NDV vectors expressing G-CSF, IL-2 or TNF resulted in decreases in tumor growth in a colon carcinoma model, and the majority of treated mice showed complete and long-lasting remission. The NDV LaSota strain showed superior suppression of tumor growth in comparison to a recombinant rNDV vector when both expressed IL-2 and IL-15 [80]. NDV-IL-15 significantly increased the levels of IFN-γ compared to NDV-IL-2. The survival rate of the NDV-IL15 group was 26.67 % higher than the rate of NDV-IL2 group [80]. Another study using recombinant NDV expressing IL-2 and TRAIL (tumor necrosis factor-related apoptosis inducing ligand) increased anti-neoplastic activity through induction of apoptosis and induced proliferation of CD4+ and CD8+ cells in mice, which reduced tumor development [83]. Prolonged survival was reported in mice carrying hepatocellular carcinoma (HCCs) and melanoma cells, which received the rNDV-IL-2-TRAIl vector [83]. In order to reduce the environmental risk of NDV, which is toxic in birds, and the neurotoxicity of VSV (Vesicular stomatitis virus), a hybrid vector was engineered, where the VSV G protein was replaced with NDV hemagglutinin-neuraminidase (HN) and further modified by fusion proteins. The hybrid demonstrated lower neurotoxicity, and no virulence in embryonated chicken eggs was detected [170].

#### 3.5.2. Clinical Trials using NDV

NDV has been applied in various clinical trials for cancer therapy. A long-term survival in phase II trials of ovarian, stomach and pancreatic cancers was observed [171]. Three hundred and thirty-five individuals with colorectal cancer were immunized with NDV vectors in a phase III trial, which provided prolonged survival and improved life quality [120]. However, in a randomized double-blind phase II/III trial in melanoma patients no significant differences between patients vaccinated with NDV and the placebo group were seen [172]. 

### 3.6. Current Stage of Melanoma Treatment Using Alphaviruses

Alphaviruses have been subjected to vector engineering procedures, including Semliki Forest virus (SFV), Sindbis virus (SIN) and Venezuelan equine encephalitis virus (VEE) [173,174,175]. RNA replicons as well as recombinant alphavirus particles and DNA/RNA layered plasmids have been applied for cancer therapy. In one study, mice were immunized with SFV-LacZ RNA, which resulted in tumor regression [58]. Since oncolytic alphaviruses occur naturally, they have been isolated and engineered from avirulent SFV strains. Using the SFV-VA7 avirulent strain, high infection rates and lysis of malignant cells were achieved, and solitary intravenous infusion prompted critical tumor relapse in a melanoma SCID mouse model [60]. 

#### 3.6.1. Preclinical Studies with Alphaviruses 

Tumor targeting represents one important consideration when using alphaviruses for cancer treatment. It has been reported that SIN particles possess natural tumor targeting. After subcutaneous administration of SIN-IL12 in mice implanted with tumor xenografts the tumor load was reduced to 6.2% of control mice [176]. On the other hand, studies on SFV particles showed no tumor targeting [101]. To address this shortcoming, liposome-encapsulated SFV-LacZ particles were engineered in order to provide tumor targeting in SCID mice after systemic delivery [177]. 

Alphavirus vectors have shown promising potential in cancer treatment because of easy and fast production of high-titer recombinant particles and the possibility of administration of transcribed RNA or layered plasmid DNA [178]. Another aspect that makes alphaviruses attractive is their capacity for extreme cytoplasmic RNA replication and transient high-level recombinant protein expression. Compared to an adenovirus vector, SFV-based HPV E6-E7 immunization required 100- to 1000-fold lower doses to evoke similar therapeutic responses in a mouse model [179]. Similarly, oncolytic alphaviruses and their tumor-targeting properties provided superior tumor growth inhibition and eradication. Another important aspect is that alphaviruses do not possess widespread immunity in human and animal populations, despite some epidemics reported for certain alphaviruses such as SFV, SIN and VEE [178]. 

Alphavirus DNA replicon vectors have been evaluated in animal models. In this context, expression of the HSV type 1 glycoprotein B from a SIN DNA vector elicited a broad range of immune responses, which included virus-specific antibodies, cytotoxic T cells and protection against lethal virus challenges in mice at a significantly lower dose compared to a conventional plasmid [180]. In another study, an SFV DNA replicon vector carrying the bovine viral diarrhea virus (BVDV) p80 (NS3) was administered into the quadricep muscles of BALB/c mice; this generated significant cytotoxic T-lymphocyte activity and cell-mediated immune responses against cytopathic and noncytopathic BVDV [181]. It was observed that, in general, lower doses (100-fold to 1000-fold) of alphavirus DNA replicon were needed to achieve the same level of response observed for conventional DNA vaccines [181,182]. 

RNA-based delivery has also proven efficient in vaccine development [58]. It has been reported that a single intramuscular injection of 0.1 µg SFV-LacZ replicon RNA generated antigen-specific antibody and CD8+ T cell responses in vaccinated animals [58]. The immunization provided protection in mice challenged with colon tumor cells. In another approach, a SIN RNA replicon expressing the rabies virus glycoprotein gene was applied for immunization of mice using only 10 µg of SIN-Rab-G RNA [183]. Compared to a conventional rabies DNA vaccine and the Rabipur commercial cell culture vaccine, similar cellular and humoral immune responses and protection against rabies were obtained.

#### 3.6.2. Clinical Trials with Alphaviruses

In attempts to improve efficacy and to provide tumor targeting, SFV particles were encapsulated in liposomes [121]. Intravenous administration of encapsulated SFV particles expressing IL-12 (LipoVIL12) enhanced IL-12 plasma levels by 10-fold in a phase I clinical trial in kidney carcinoma and melanoma patients. The encapsulation enhanced tumor targeting as well as prevented recognition by the host immune system, which allowed re-administration without evoking immune responses against SFV [121]. Additional clinical trials have been conducted with alphaviruses for various cancer types and viral diseases, demonstrating good safety of the immunization process, although further dose optimization is still required. This represents a promising aspect for further development of alphavirus applications in melanoma treatment.

## 4. Conclusions and Future Aspects

In summary, oncolytic viruses have proven powerful in several preclinical tumor models and clinical studies. They have provided suppression of tumor growth and regression as well as increased quality of life in patients (Table 2). The range of oncolytic viruses (adenoviruses, AAV, HSV, Coxsackievirus, NDV, retroviruses, etc.) allows modifications related to expression, packaging and host range tropism. However, further improvements in dose optimization and therapeutic efficacy are needed, especially in the area of melanoma therapy. Today, many virus vectors have demonstrated promising efficacy in combination with standard melanoma therapy, including immunotherapy, either in the form of MEK and BRAF inhibitors or in combination with chemotherapy (dacarbazine and cisplatin). It seems that no single-virus vector system shows superiority and, therefore, development of different expression systems in parallel is recommended. In conclusion, advances in gene therapy and the approval of several drugs will contribute to virus-based therapy potentially becoming a key asset in modern clinical applications, including related to malignant melanoma. 

## Figures and Tables

**Table 1 biomedicines-08-00060-t001:** Viruses and their vectors applied in overall gene therapy [36].

Virus	Genome	Characteristics	Cancer Types
Adenovirus Ad5	dsDNA	Strong immunogenicity Broad host range [37,38]	Brain [39] Colon [40,41] Esophaegal [42] Gastric [43] Liver [44] Lung [45]
Adeno associated viruses AAV (2, 3, 5, 6, 8, 9)	ssDNA	Slow expression onset Chromosomal integration [46,47,48,49]	Brain [50] Breast [51] Liver [52,53] Lung [54] Prostate [55] Retinoblastoma [56]
Alphaviruses SFV, SIN	ssRNA	Broad host range low immunogenicity [32]	Brain [57] Colon [58] Lung [59] Melanoma [60,61] Osteosarcoma [62] Pancreas [63] Prostate [64]
Herpes simplex virus HSV1, HSV	dsDNA	Broad host range Latent infection [65,66]	Brain [67,68] Colon [69] Prostate [70] Sarcoma [71] Skin [72]
Reoviruses Reolysin	dsDNA	Specific replication in transformed host cells Non-pathogenic in human Anti-cancer activity	Melanoma [73]
Retroviruses MMSV MSCV	ssRNA	Random integration Long-term expression [74,75,76]	Glioma [77,78]
Lentiviruses HIV-1, HIV-2	ssRNA	Broad host range Low cytotoxicity [76,79,80]	Breast [81] Gastric [82] Liver [83] Pancreas [84] Leukemia [85]
Rhabdoviruses Rabies, VSV	ssRNA	Low immunogenicity High transient expression [86]	Sarcoma [86]
Measels virus	ssRNA	Oncolytic strains [87]	Breast [87] Liver [88]
Newcastle disease virus	ssRNA	Improved oncolytic virus [89,90]	Melanoma [91] Lung [92] Liver [93,94]
Picornaviruses	ssRNA	Oncolytic strains [95]	Melanoma [96] Breast [97,98] Prostate [99]
Poxviruses	dsDNA	Improved oncolytic virus [100,101,102]	Pancreas [103] Prostate [104] Colon [105] Glioblastoma [106]

**Table 2 biomedicines-08-00060-t002:** Clinical trials of melanoma treatment using oncolytic viruses.

Virus	Trial	Clinical Results	References
HSV-1	Phase III	Improved durable response rate (16.3% vs. 2.1%), overall response rate (26.4% vs. 5.7%) and longer median survival (patients with non-surgically resectable melanoma) [116]	[116,117]
HSV-1	Phase Ib/II	50% objective response rate 44% of patients had a durable response lasting >6 months	[118]
HSV T-VEC	Phase II/IIIb	Overall survival superior for patients with stage III and IV M1a melanoma Significant clinical benefits in phase III	[114]
Reovirus	Phase II	No objective responses seen	[73]
Coxsackievirus	Phase Ib	Stable disease in 26.7% of patients Best ORR of 60%	[119]
Coxsackievirus	Phase II	Durable responses in melanoma metastases (injected and uninjected)	[116]
Newcastle Disease Virus	Phase II/III	No superiority compared to controls	[120]
Alphaviruses	Phase I	10-fold enhanced IL-12 plasma levels Encapsulation enhanced tumor targeting	[121]

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
