# Peer review of "Viral Vector-Based Melanoma Gene Therapy"

_biomedicines, 2020, doi:10.3390/biomedicines8030060_

Round 1

Reviewer 1 Report

The manuscript by Hromic-Jahjefendic A. and Lundstrom K., elaborates on gene therapy approaches in the treatment of a skin cancer - melanoma. Because the disease affects approximately 5% of the world’s population, it constitutes an important health problem with vast socio-economic implications. Thus, the topic can potentially attract a broad readership.

The manuscript is generally well written, with a brief introduction of various treatment options that are currently available to patients with melanoma. The authors put great emphasis on clinical trials with medicinal products based on viral vectors, which are well described in the article and supported by significant references.

However, the articles has also some limitations, which are listed below:

  1. Since the article is focused only on melanoma, it should provide more information about epidemiology or statistics of melanoma occurrence worldwide, in the introduction section.
  2. Table 1 from this article repeats the information from a different article of the same author (Lundstrom K. Viral Vectors in Gene Therapy. Diseases. 2018;6(2):42), only with extension of references for different cancer types. Please, modify.

Minor point:

  1. Abbreviations: CR, PR, SD from the lines 270-271 have already been explained on a previous page- lines: 204-205 and should not be explained for the second time.

Author Response

  1. Since the article is focused only on melanoma, it should provide more information about epidemiology or statistics of melanoma occurrence worldwide, in the introduction section.

R: Additional information is added on epidemiology and statistics.

  1. Table 1 from this article repeats the information from a different article of the same author (Lundstrom K. Viral Vectors in Gene Therapy. Diseases. 2018;6(2):42), only with extension of references for different cancer types. Please, modify.

R: We disagree as there are certainly differences between Table 1 and the table in a previous publication and these are facts, which we find difficult to present in a different way. In Table 1 is a description of cancer types and the column of packaging capacity is not included. Moreover, the list of virus vectors is different.

Minor point:

  1. Abbreviations: CR, PR, SD from the lines 270-271 have already been explained on a previous page- lines: 204-205 and should not be explained for the second time.

R: Correction has been made.

Reviewer 2 Report

This is a very well written review article with detailed explanation of the disease condition, current status of treatment modalities, supporting references, regulatory prospectives and authors views on Viral Vector-based Gene Therapy for Melanoma.

One minor suggestion: Why authors did not discuss in a section about retro/lentivurises based approaches for meloanoma, though authors well listed and referenced in Table-1? Below are certain related refereces on Gene Therapy for Melanoma:

1) DOI: 10.5772/54936

2) https://doi.org/10.1002/stem.170191

3) https://doi.org/10.3389/fmicb.2018.01448

Although a section on retro/lentiviruses is critically not required in the current manuscript, but it may be helpful to the readers since this is a review article.    Overall nicely done!  

Author Response

Although a section on retro/lentiviruses is critically not required in the current manuscript, but it may be helpful to the readers since this is a review article.    Overall nicely done!  

R: A section on retroviruses/lentiviruses has been added.

Reviewer 3 Report

 in this short review, Jahjefendic et al summarized the achievements of preclinical and clinical studies using viral vectors with the focus on malignant melanoma cancer therapy. Overall the review is good and it shall be published as it is. However, the authors can add more up to date scientific advancements with supportive references and diagrams in this field.

Author Response

In this short review, Jahjefendic et al summarized the achievements of preclinical and clinical studies using viral vectors with the focus on malignant melanoma cancer therapy. Overall the review is good and it shall be published as it is. However, the authors can add more up to date scientific advancements with supportive references and diagrams in this field.

R: Additional information on retroviruses/lentiviruses has been included.